# Extracellular Vesicles in Innate Immune Cell Programming

**DOI:** 10.3390/biomedicines9070713

**Published:** 2021-06-23

**Authors:** Naveed Akbar, Daan Paget, Robin P. Choudhury

**Affiliations:** 1Radcliffe Department of Medicine, University of Oxford, Oxford OX3 9DU, UK; daan.paget@lmh.ox.ac.uk (D.P.); robin.choudhury@cardiov.ox.ac.uk (R.P.C.); 2Department of Pharmacology, University of Oxford, Oxford OX1 3QT, UK

**Keywords:** exosomes, transcription, neutrophil, monocyte, hematopoietic stem cell

## Abstract

Extracellular vesicles (EV) are a heterogeneous group of bilipid-enclosed envelopes that carry proteins, metabolites, RNA, DNA and lipids from their parent cell of origin. They mediate cellular communication to other cells in local tissue microenvironments and across organ systems. EV size, number and their biologically active cargo are often altered in response to pathological processes, including infection, cancer, cardiovascular diseases and in response to metabolic perturbations such as obesity and diabetes, which also have a strong inflammatory component. Here, we discuss the broad repertoire of EV produced by neutrophils, monocytes, macrophages, their precursor hematopoietic stem cells and discuss their effects on the innate immune system. We seek to understand the immunomodulatory properties of EV in cellular programming, which impacts innate immune cell differentiation and function. We further explore the possibilities of using EV as immune targeting vectors, for the modulation of the innate immune response, e.g., for tissue preservation during sterile injury such as myocardial infarction or to promote tissue resolution of inflammation and potentially tissue regeneration and repair.

## 1. Extracellular Vesicles

Extracellular vesicles (EV) are membrane-enclosed lipid envelopes that can mediate cellular communication locally in tissue microenvironments or upon liberation into peripheral blood [1,2,3,4,5,6,7], lymph [8] and cerebrospinal fluid [9] remotely from their site of origin. EV bear features of their parent cells such as the presence of proteins, lipids and nucleic acids (mRNA, microRNAs (miRNAs), long non-coding RNAs and DNA), which are capable of initiating cellular signalling in recipient cells [10]. Importantly, EV from a range of pathologies which have a strong inflammatory component such as infection, rheumatological diseases, cardiovascular diseases (including acute myocardial infarction, stroke and atherosclerosis), cancer and metabolic disorders such as obesity and diabetes show alterations in EV size, concentration, cargo and show effects on innate immune cells [11,12,13]. Innate immune cells are the principal effector cells in the clearance of EV from the systemic circulation and they show predilection for EV uptake [14] compared with other cells, such as lymphocytes [15,16].

Neutrophils and monocytes are present in peripheral blood and are recruited to sites of infection, injury and inflammation. Mechanisms of immune cell recruitment to tissues and the factors that enable their trafficking and infiltration are well documented [17,18]. Conventionally, innate immune cells patrol the systemic vasculature and undergo inflammatory activation in response to local signalling, manifesting a broad repertoire of pro-inflammatory and inflammation-resolving responses. In addition to the circulating pool, reserves of cells in the bone marrow and the spleen contribute large numbers of immune cells to inflammation-related pathologies [19]. Acutely after sterile injury, such as myocardial infarction, the spleen deploys large numbers of monocytes to the peripheral blood, which mediate further injury to the heart [20]. Similarly, the spleen deploys neutrophils following infection [21]. The functional characteristics of these peripheral blood innate immune cells and those in the splenic and bone marrow reserve are becoming clearer though use of multi’-omics’ techniques. Single cell-RNA-sequencing (sc-RNA-seq) analyses have identified sub-sets of cells that perform important functions in myocardial injury [22,23,24] and tissue repair [25,26], potentially opening opportunities to modulate these cells therapeutically. 

We have demonstrated a role for endothelial cell derived EV in innate immune cell mobilisation from the splenic reserve and in the transcriptional programming of monocytes and neutrophils in the blood, prior to recruitment to tissues [7,27]. Similar transcriptional programming prior to tissue consignment is reported following viral infection e.g., SARs-CoV-2, brain aneurysms and the peripheral monocyte pool shows phenotypic trained immunity following inoculation [28,29,30]. This demonstrates that innate immune cells in a broad range of conditions show transcriptional alterations prior to tissue recruitment. Viral and inoculation programming and training of innate immune cell may be mediated in the peripheral blood by viral components but the manifestation of these transcriptome alterations in sterile tissue injury are poorly described. EV are a ubiquitous means of cell communication, which impact broadly on innate immune function [31]. Here, we focus on the opportunities for immunomodulation of neutrophils, monocytes and their pre-cursor haematopoietic stem and progenitor cells (HSCs) by EV (Figure 1). A better understanding of how immune cells are transcriptionally activated and programmed *prior to* tissue recruitment by EV and during cellular differentiation could allow the development of targeted therapeutic strategies, e.g., to promote tissue resolution of pro-inflammatory signals and induce tissue repair and regeneration. There is considerable crosstalk between cancer and tumour derived EV, HSCs and their progeny but these have been recently reviewed and will not be discussed in detail here [32]. 

## 2. Neutrophils 

Neutrophils are the most abundant leukocyte population in peripheral blood and arrive rapidly at sites of tissue inflammation, infection and injury [33,34,35,36]. Neutrophil functions were thought to be limited due to a diminished transcriptional capacity in mature cells and short lifespans [36]. New single cell (sc)-RNA-sequencing data show diverse transcriptional heterogeneity in neutrophils [35] and studies have reported lifespans of up to a week in vivo [36]. Phagocytosis of apoptotic neutrophils is known to induce anti-inflammatory responses in macrophages and mediates tissue repair and healing in vivo following myocardial infarction [37]. These anti-inflammatory effects are partially mediated by neutrophil proteins such as interleukin-10 (IL-10) [37] and by neutrophil derived EV [38,39,40,41,42,43]. Neutrophils can release at least two sub-classes of EV, termed: neutrophil derived trails (NDTRS), which are generated by integrin mediated interactions by migrating neutrophils in response to vascular wall forces and neutrophil derived microvesicles (NDMV), which are dependent on the PI3K pathway and released by membrane blebbing following neutrophil activation [41,42]. NDTRS and NDMV share common proteins but differ in their inflammatory and anti-inflammatory functions, respectively [44]. Therefore, we will focus on the role of NDMV in cellular programming (Figure 2).

NDMV express EV marker CD63, adhesion molecules CD11b, CD18, CD62L, the complement receptors CR1, CD55, granule proteins myeloperoxidase (MPO) and lactoferrin, which enable receptor mediated signalling in recipient cells. NDMV are present in the plasma of healthy individuals and show augmentation in patients with sepsis [43] and chronic obstructive pulmonary disease (COPD) [45]. NDMV reduce reactive oxygen species (ROS) production in neighbouring neutrophils, leading to an anti-inflammatory phenotype [44]. In addition, NDMV target NF-ϰB transforming growth factor β-activated kinase 1 (TAK1) and IϰBβ activation in macrophages [46]. Inhibited NF-ϰB translocation to the nucleus limits tumour necrosis factor (TNF), IL-6, IL-8 [46,47] and modulates autophagy in macrophages [40]. Some of these responses are in part mediated by NDMV phosphatidylserine and annexin-A1 [40,46]. However, NDMV are diverse and their cargo include miRNAs, such as miR-126, miR-150 and miR-451a, which similarly mediate anti-inflammatory responses in recipient cells such as monocytes [41].

NDMV can also modulate the phenotype of endothelial cells by regulating endothelial cell permeability and inducing endothelial cell dysfunction [48,49]. NDMV alter endothelial cell tight junctions [48] and enhance the formation of atherosclerotic plaques by the transfer of miR-155 to mediate macrophage accumulation [49]. Whereas mesenchymal stem cell (MSC) EV modulate neuroprotection during ischemic injury by inhibiting neutrophil recruitment and mediate similar protective effects to those observed with neutrophil depletion [50]. Whereas, cancer derived EV enhance neutrophil survival [51] and polarise neutrophils into an anti-inflammatory N2 tumorigenic phenotype [52]. Tumour derived EV enable cancer progression by advocating a pro-metastatic microenvironment and by facilitating the recruitment of neutrophils [51,52]. Tumour derived EV activate neutrophils to induce systemic complications such as thrombus formation, which in cancer patients is associated with worse prognosis. In a mouse model of mammary carcinoma EV elevate blood neutrophil number and induce systemic neutrophil activation and release of neutrophil extracellular traps (NETs) resulting in arterial thrombi [53]. The pro-tumourgenic effect of cancer EV on neutrophils supports the possibility of targeting neutrophils with EV as immunomodulators to retard the progression of cancers [51,52,53] and possibly other diseases with an inflammatory component, where neutrophil activation contributes to pathological processes. But it remains unknown whether NDMV-protein or NDMV-miRNAs are more potent in modulating monocyte and macrophage inflammatory functions and why NDMV mediate opposing inflammatory and anti-inflammatory effects in different cells (monocytes/macrophages, endothelial cells and MSC). A better understanding of these cell type dependent effects of NDMV may enable the generation of therapeutic EV for immunomodulation of monocytes, macrophages, and endothelial cells.

Neutrophils show activation in a broad range of pathologies and NDMV diversity is in part reflective of the parent cell [41,42]. NDMV display different proteome characteristics and functional properties [54] dependent on whether neutrophils are in suspension or in contact with other cells such as endothelial cells. Peripheral blood neutrophils show significant heterogeneity and sc-RNA-sequencing studies have identified at least 8 different neutrophil sub-populations in the circulation [35]. 

Importantly, neutrophils sub-types display specific phenotype-dependent behaviour. Neutrophils undergo classical transendothelial migration and exhibit reverse transendothelial migration (rTEM) properties [54]. rTEM neutrophils differ in their expression of intracellular adhesion molecule-1 (ICAM-1) (high) and CXCR1 (low) [55]. Specific neutrophil functions, such as rTEM may influence the type of NDMV generated by neutrophils locally in tissues microenvironments versus those present in the peripheral blood. This raises the questions as to whether neutrophils in circulation release similar NDMV to tissue recruited neutrophils, those neutrophils which undergo rTEM and whether tissue NDMV are detectable in peripheral blood. An ability to isolate specific NDMV sub-populations, which represent tissue microenvironments may enable disease diagnostics and potentially patient stratification for therapy.

## 3. Monocytes and Macrophages 

Monocytes are produced in the bone marrow by pluripotent stem cells and follow a programme of differentiation from monoblasts, promonocytes and mature into monocytes before release into the systemic circulation [56]. However, in mouse models, extramedullary myelopoiesis at sites of EV clearance such as the spleen [57] also contribute large numbers of inflammatory cells following infection and sterile ischemic injury such as a heart attack. 

Acutely after sepsis, monocyte-derived EV induce functional impairment/immune-paralysis in recipient monocytes, by inhibiting the release of TNF-α [58,59]. Monocyte-derived EV carry classical monocyte-markers CD14, CD4, CD16, CD163 and CCR5 [60] but not generalised EV marker CD63 [61,62] (Figure 3). Conversely to septic challenges, exposure of human monocytes to alcohol enhances EV release, which stimulates naïve monocytes into an M2-anti-inflammatory phenotype, through increased expression of mannose receptor CD206 and scavenger receptor CD163 via EV transfer of miRNA-27a [63]. Peripheral blood monocyte-derived EV associate with high mortality rates in patients with liver cirrhosis and sepsis [58].

Monocytes accumulate EV from different cell types that can reprogram monocyte functions. Cardiac adherent proliferating cell EV are readily internalised by CD14+ monocytes and modulate expression of HLA-DR, CD86 and increased expression levels of CD206, programmed death ligand-1 (PDL-1) and release more IL-1Rα [64]. MSC-derived EV modulate monocyte phenotypes in a model of airway inflammation from a classical/inflammatory phenotype to a non-classical anti-inflammatory phenotype [65,66] and under hypoxic conditions MSC-derived EV regulate macrophage anti-inflammatory phenotype via EV miR-21-5p and promote lung cancer development [66]. MSC-derived EV fused with monocyte-derived EV show enhanced propensity to accumulate in the injured heart following myocardial infarction, induce endothelial cell maturation during angiogenesis and impact macrophage phenotypes [67]. 

EV derived from quiescent endothelial cells inhibit monocyte activation by transfer of EV-miRNA-10a, which targets NF-ϰB signalling and the inflammatory transcription factor interferon regulatory factor-5 (IRF5) [60]. Conversely, EV derived from endothelial cells in a pro-inflammatory context mediate monocyte mobilisation from the spleen through enrichment of the endothelial cell-EV-miRNA-126 and induce macrophage activation in the liver [2]. Similarly, hepatocyte derived EV induce the recruitment of monocyte-derived macrophages and mediate inflammatory injury in a model of diet-induced steatohepatitis [68] and in a model of high fat, fructose and cholesterol feeding by the action of hepatocyte EV-integrin β1 (ITGβ1), which induced the adhesion of monocytes to the sinusoidal endothelium [69].

Upon tissue recruitment peripheral blood monocytes can differentiate into macrophages. Monocyte-derived macrophages release small EV that are positive for EV markers ALIX, CD63 and CD81, which accumulate rapidly (< 3 h) in recipient macrophages [70] and induce differentiation of other macrophage via transfer of EV-miR-223 [71]. Macrophages internalise EV from a wide variety of sources including breast milk derived EV [72], tumour derived EV, which activate macrophages [73] and acute myeloid leukaemia (AML) derived EV, which induce a myeloid derived suppressor cell (MDSC) differentiation phenotype in recipient macrophages [74]. Macrophages are the principal effector cell in EV clearance in vivo and depletion of macrophages using clodronate liposomes attenuates EV-capture and EV-mediated macrophage activation [75,76].

Monocyte-derived macrophages challenged with a variety of stimuli including chemokines show differential packing of EV-miRNAs [77] that induce integrin expression in endothelial cells [78], in adventitia vascular smooth muscle cells they elevate matrix metalloproteinases-2 (MMP-2) through JNK and p38 to promote aneurysms [79] and induce proneural-to-mesenchymal transition (PMT) in glioblastoma by transfer of EV- miR-27a-3p, miR-22-3p, and miR-221-3p [80]. 

In a model of doxorubicin toxicity cardiac EV treated with doxorubicin showed greater preponderance for macrophage activation than doxorubicin treated hepatocyte derived EV, which did not activate macrophages [81]. It remains unclear whether these differential effects of doxorubicin induced EV on macrophage activation are due to cell type difference and/or difference in EV-cargo, i.e., the presence of particular EV-proteins or EV-miRNAs. 

Macrophages are known for their role in infection and macrophages infected with pathogens such as *Mycobacterium tuberculosis* produce and release EV, which show differential membrane protein enrichment [82]. Similarly, human macrophages infected with *Mycobacterium bovis*, Bacille Calmette-Guérin (BCG) release EV with enrichment for 11 EV-miRNAs, which target cellular metabolism and energy related pathways [76], possibly to stimulate metabolic and inflammatory responses in neighbouring cells. Macrophages infected with *Mycobacterium tuberculosis* produce and release two distinct populations of EV, one from the host macrophage, that are positive for CD9, CD63 and the other EV population contains *Mycobacterium tuberculosis* molecules, including lipoglycans and lipoproteins [40,82,83]. This demonstrates a pathway, where living bacteria inside infected macrophages release EV containing microbial components and initiate TLR-2 activation and cytokine release in naïve uninfected macrophages [84]. A sequential combined ‘*-omics*’ analysis of both EV-protein and EV-RNA in the same infected macrophages could identify infection-specific EV-signatures and aid the understanding of how pathogens impact macrophage EV biogenesis, release and alter EV-cargo. Macrophages infected with *Salmonella typhimurium* can package and deliver LPS to naïve macrophages, initiating TLR-4-inflammatory cytokine release, which may stimulate further recruitment of monocytes and induce their proliferation [84]. Gram-negative bacteria are able to induce the activation of caspase-11 in the cell cytosol through generation, release and uptake of EV but these bacteria themselves do not enter the cytosol [85]. Similarly *Leishmania* transfer pathogenic EV-proteins to host cells to mediate macrophage activation [86]. Additionally, monocyte-derived-macrophages can load cholesterol and lysosomal β-hexosaminidase enzymes into EV [87]. The ability of pathogens to release EV within host cells and alter their composition opens the possibility of targeting EV-pathways to prevent the incorporation of specific proteins, peptides or RNAs into EV for therapeutics, perturbing infection and inflammation.

Macrophage derived EV show distinct profiles depending on the activation status of the cell [88]. The pro-inflammatory effects of macrophage EV derived from pathogen infected cells and the anti-inflammatory effects of monocyte-derived EV from alcoholic patients, from apoptotic cells [89], adipocytes [6], MSC and doxorubicin stimulated cells suggests dynamic properties of monocyte and macrophages EV [90]. Therefore, monocyte-derived EV may provide utility as diagnostic biomarkers for the assessment of pathologies where monocyte phenotypes contribute to the inflammatory disease such as infection, dyslipidaemia, diabetes, obesity and cardiovascular diseases. It is unclear whether EV derived from the same cell type, i.e., monocyte EV delivered to other monocytes or whether neutrophil or other cell derived EV delivered to monocytes mediate greater alterations in the recipient cell’s function. A better understanding of these inter-cell EV communications is necessary to therapeutically capitalise on opportunities to perturb inflammatory signalling and to utilise therapeutic-EV for immunomodulation.

A number of monocyte EV investigations have utilised monocyte-cell lines such as THP-1 monocytes or THP-1 monocyte-derived macrophages to investigate EV release and macrophage activation [91,92]. These studies report distinct effects derived from macrophage EV but not monocyte-derived EV. THP-1 monocyte-derived macrophage culture requires inflammatory agonists such as phorbol 12-myristate 13-acetate (PMA) for cellular differentiation, which simultaneously polarises cells to a predominantly pro-inflammatory phenotype [93]. Immortalised cell lines do not accurately represent normal cellular functions and responses, such as those obtained from primary peripheral blood cells from healthy volunteers, patients or rodents. Immortalised cells bare a predominantly pro-inflammatory phenotype, which may limit the translation of many studies into diagnostic and therapeutic strategies [94]. Detailed studies utilising primary human peripheral blood monocytes would enable a better understanding of how monocytes release EV in response to specific agonists, which may enable the identification of EV-components for diagnostics and/or as therapeutic targets. 

A better understanding of how EV mediate monocyte interactions to modulate monocyte and macrophages motility, gene expression and monocyte function will enable the generation of EV-therapeutics; to potentially modulate monocyte and macrophage functions following infection, tissue injury, tissue inflammation and to potentially modulate monocyte-macrophage activation towards tissue repair and regeneration.

## 4. Bone Marrow Programming

The majority of in vivo innate immune EV studies are conducted in the mouse and may not translate to humans due to inherent species differences in cellular signalling. However, HSCs populations within the bone marrow show high similarity between the mouse and the human [95]. HSCs are sensitive to external stimuli and alterations in the gut microbiome [7], oxygen [96,97] and septicaemia [98] can re-programme cells and alter their cellular functions. 

EV mediate direct and indirect effects on both the bone marrow stromal cells and HSCs. The bone marrow HSCs receive a multitude of signals from the systemic circulation to regulate haematopoiesis [99]. LPS challenged monocyte-derived EV can regulate cytokine gene expression in MSC and induce an immunomodulatory phenotype by targeting CXC chemokines, IL-1 and by upregulating genes for matrix metalloproteinase-1 (MMP-1) and MMP-3 [100]. Monocyte-derived EV during sepsis induce bone marrow hyperplasia and leukocytopenia [58].

MSC release EV that induce the regenerative expansion of HSCs through loss of quiescence by stimulating MyD88-TLR-4 signalling cascade dependent on NF-ϰB-HIF-1α [101,102]. Mobilisation of HSCs from the bone marrow to the peripheral blood can be induced by administration of granulocyte colony-stimulating factor (G-CSF), which lowers vascular cell adhesion molecule-1 (VCAM-1) expression in the bone marrow through accumulation of EV-miRNA-126 [103]. Similarly pancreatic cancer derived EV bind to CD11b+ bone marrow cells and downregulate genes for monocyte and macrophage: activation, trafficking and expression of inflammatory molecules [104]. 

Rab proteins are upregulated in cancers and increase EV release [105]. Cancer EV are delivered to the bone marrow from the systemic circulation can re-programme and mobilise bone marrow cells, to favour a pro-vasculogenic and pro-metastatic niche formation phenotype through the receptor tyrosine kinase MET [106]. A melanoma-specific EV-protein signature comprised of tyrosine-related protein-2 (TYRP2), which predicts disease progression in melanoma patients [106], very late antigen-2 (VLA-2), heat shock protein 70 (HSP70), an isoform of HSP90 and the MET oncoprotein promotes this vasculogenic phenotype. Rab27α RNAi lowers EV-production and prevents cancer-EV from programming bone marrow cells to a pro-vasculogenic phenotype [107]. Rab27 α/β double knock-out mice have impaired EV release and show a hyper-inflammatory phenotype, which can be dampened by treating the mice with HSC derived EV rich in miR-155 [107]. AML derived EV enter HSCs and alter protein synthesis via EV-miRNA-1246 inducing ribosomal protein S6 hypo-phosphorylation and elicit DNA breaks [108].

These EV mediated effects on bone marrow and stromal cells alter immune cell proliferation and the capacity of differentiating cells to respond to physiological and pathological signals. NDMV contain S100A8, S100A9, which can induce granulopoiesis [53] and may mediate emergency haematopoiesis during sterile injury and infection.

MSCs derived EV from healthy donors administered to a patient to prevent graft-versus-host disease showed a reduction in clinical symptoms, tolerance of the mesenchymal stem cell derived EV and a reduction in cytokines [109]. 

## 5. Immunometabolic Reprogramming 

Numerous cell types, including neutrophils, monocytes and HSCs undergo metabolic shifts during cellular differentiation, cellular activation and in response to perturbations in nutrients including elevations in glucose [110,111]. Shifts in cellular metabolic status can alter EV production and cargo [112,113] but it remains unknown how the metabolic profile of immune cells and vascular lumen endothelial cells, which provide an interface with circulating peripheral blood immune cells and underlying tissues, alter EV-pathways to promote disease associated signalling.

Cancer cell derived EV can be internalised by stromal cells and induce a metabolic shift by lowering mitochondrial respiration to oxidative phosphorylation to meet energy requirements, increasing lactate production, which may modify the local tissue environment to support the growth of cancers in a reverse Warburg effect [114]. The intracellular events leading to cancer-EV reprogramming of HSCs remains unknown. EV carry inflammatory cytokines, metabolites, lipids and RNA, which may mediate reprogramming of HSCs through a number of mechanisms including modulation of their metabolism [115]. 

A better understanding of the balance between inflammatory cell phenotype and immune cell metabolism opens the possibility of targeting immune cells to re-programme their cellular function. By favouring tissue reparative neutrophil-N2 or macrophage-M2-phenotypes, possibly through therapeutic bioengineered EV that bear specific metabolites, proteins and RNA, it may be possible to dampen inflammation and promote immune mediated tissue repair and regeneration. Combined ‘*-omics*’ approaches to investigate how EV modulate cellular metabolism, the epigenome and gene expression profiles in recipient cells may open novel therapeutic strategies for immune cell modulation.

## 6. Conclusions and Perspectives

EV number, size and their biologically active material is altered in numerous inflammatory conditions and EV can alter the cellular functions of neutrophils, monocytes, macrophages and their precursor HSCs. Innate immune derived EV show diversity in both their EV-associated protein markers and those protein markers associated with their lineage [116].

For EV to serves as specific, reliable and effective disease biomarkers in depth screening must be undertaken to identify specific features that relate EV characteristics to disease states, clinical correlates and outcome, i.e., mortality. Studies should capitalise on the use of human samples, disease models and in vitro investigation to dissect the nature of the EV phenotype and explore avenues to limit potentially pathological signalling. A better understanding of how EV signal in a range of clinical manifestations will allow the development of both disease specific EV-diagnostic tests and EV-based-therapeutics.

Although a range of diagnostic tests provide information on systemic changes in tissue homeostasis and cellular response, soluble biomarkers often fail to discriminate their source and do not provide insight into the status of individual tissues. EV carry biologically active material from their parent cell, which many include RNA/DNA and/or proteins that are associated with distinct cell types. For instance, miRNA-126 and the integrin-VCAM-1 are associated with endothelial cells. 

Methods for detecting low abundant malignant tumour cells and soluble blood borne factors are advancing, to allow proteomic, metabolic and RNA-sequencing analysis of small and limited amounts of patient material including biopsy samples. Similar on-chip technologies may allow the rapid isolation and characterisation of EV for diagnosis of pathology or to monitor disease progression in response to therapeutic intervention. The development of on-chip-EV based diagnostics may mitigate the need to obtain repeated biopsies from patient’s overtime and will allow automation of EV based diagnostic tests.

In depth characterisation of EV, potentially time-resolved through the development of subclinical pathology, presentation and after clinical intervention will provide data important for patient stratification/precision medicine. This will be coupled with better understanding of how EV-biogenesis and cargo are altered in specific cell types and in response to environmental and physiological stimuli to aid the development of therapeutics, possibly through the generation of bioengineered EV. Recent reports highlight the ability to incorporate exogenous mRNA into endogenous EV by utilising lipid nanoparticles at specific molar ratios. This approach opens an exciting possibility to generate EV, which carry specific mRNAs [117]. Therapeutics that modify vesicle biogenesis and/ or modulate the incorporation of biologically active material such as lipids, RNA and DNA may provide clinical utility to perturb pathological signalling.

## Figures and Tables

**Figure 1 biomedicines-09-00713-f001:**
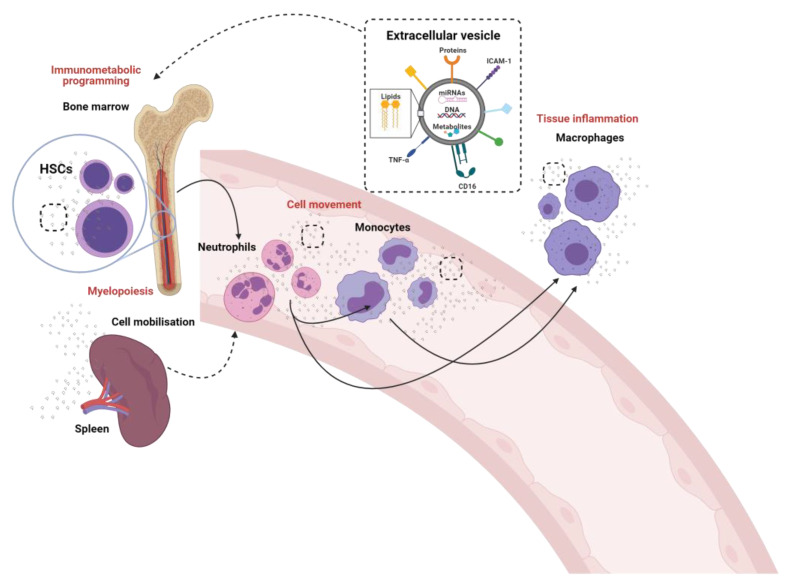
The crosstalk between innate immune cells through the generation and release of extracellular vesicles. The known (solid lines) and proposed (dotted lines) role of extracellular vesicles in the modulation of neutrophils, monocytes and macrophages and their pre-cursor hematopoietic stem cells (HSCs).

**Figure 2 biomedicines-09-00713-f002:**
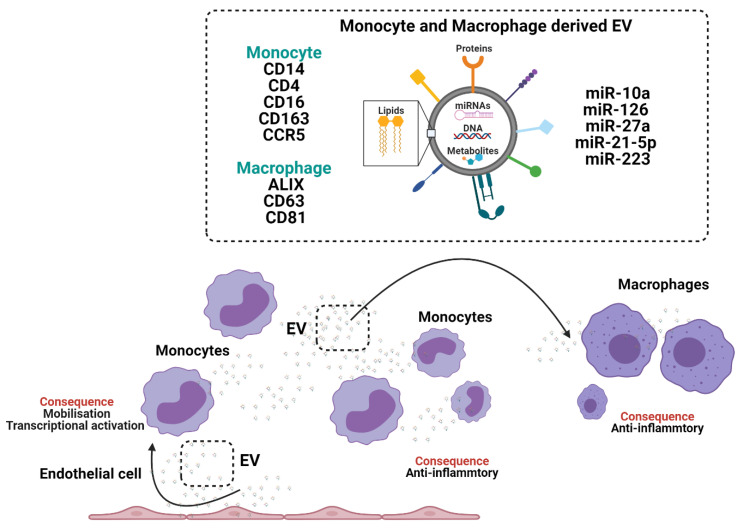
The crosstalk between neutrophils, monocytes and macrophages through the generation and release of extracellular vesicles (EV). Neutrophil derived microvesicles (NVDMs). Myeloperoxidase (MPO).

**Figure 3 biomedicines-09-00713-f003:**
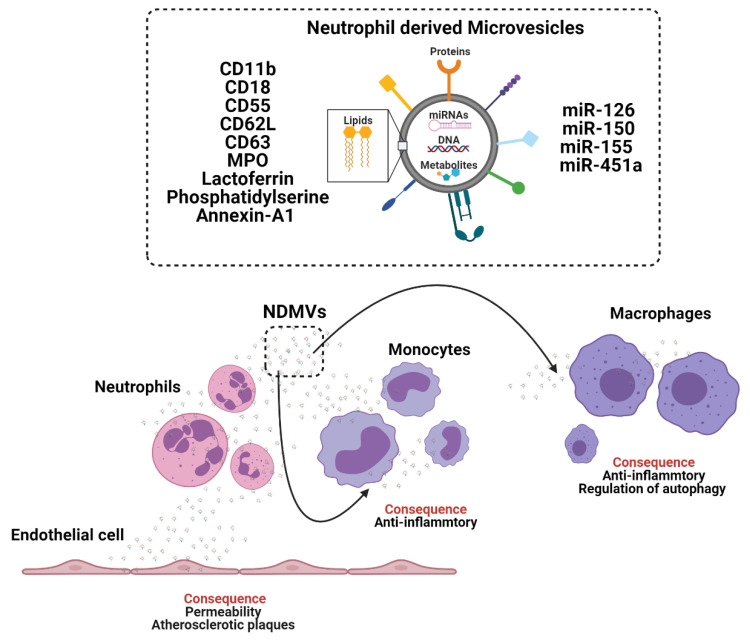
The crosstalk between monocytes and macrophages through the generation and release of extracellular vesicles (EV).

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
