# Peer review of "Extracellular Vesicles in Innate Immune Cell Programming"

_biomedicines, 2021, doi:10.3390/biomedicines9070713_

Round 1

Reviewer 1 Report

In this manuscript, the authors reviewed the recent progress regarding the critical role of a broad repertoire of extracellular vesicles (EVs) produced by MSCs, endothelial cells, monocytes, macrophages, neutrophils, and HSCs in regulating the innate immune system. They also discussed how pathological conditions such as infection and MI regulate the component of EVs. Finally, they discussed the translational potential to modulate innate immune responses by targeting EVs. Overall, the review is interesting, up-to-date, and well written. The only concern is that the authors mainly summarized the crosstalk between EVs and the innate immune system in cardiac vascular diseases, whereas the interaction between cancer cell-derived EVs and myeloid cells is largely ignored. It is now well accepted that cancer cell-derived EVs reprogram the differentiation of tumor-associated macrophages and dendritic cells to evade immune surveillance. Following are my specific comments:

  1. The authors should include a more detailed figure to illustrate the interaction between EVs and innate immune cells;
  2. The consistency of abbreviations. For instance, the authors use EVs and EV for extracellular vesicles randomly. 
  3. Page 3, paragraph 3, line 5: change NMDV to NDMVs;
  4. Page 3, last paragraph: change neutrophils to neutrophils'
  5. Page 4: change MSCs derived EV to MSC-derived EVs, monocyte derived macrophages to monocyte-derived macrophages.

Author Response

We thank the reviewers  and editors for their appraisal and careful consideration of our review manuscript. We are pleased to address all comments raised and include a point-by-point rebuttal below.

Comments and Suggestions for Authors

In this manuscript, the authors reviewed the recent progress regarding the critical role of a broad repertoire of extracellular vesicles (EVs) produced by MSCs, endothelial cells, monocytes, macrophages, neutrophils, and HSCs in regulating the innate immune system. They also discussed how pathological conditions such as infection and MI regulate the component of EVs. Finally, they discussed the translational potential to modulate innate immune responses by targeting EVs. Overall, the review is interesting, up-to-date, and well written.

We thank the reviewer for their careful consideration of our review manuscript.

The only concern is that the authors mainly summarized the crosstalk between EVs and the innate immune system in cardiac vascular diseases, whereas the interaction between cancer cell-derived EVs and myeloid cells is largely ignored. It is now well accepted that cancer cell-derived EVs reprogram the differentiation of tumor-associated macrophages and dendritic cells to evade immune surveillance.

We are pleased to include some additional discussion on cancer derived EVs in myeloid cells programming. However, this topic has been extensively reviewed recently by a number of publications thus we have highlighted this in our revised manuscript and cited relevant articles.

Following are my specific comments:

  1. The authors should include a more detailed figure to illustrate the interaction between EVs and innate immune cells;

We have included two additional Figures in the revision, which contain more detail on how EV-programme innate immune cells.

  1. The consistency of abbreviations. For instance, the authors use EVs and EV for extracellular vesicles randomly. 

We have been through the manuscript carefully and altered our abbreviations for consistency.

  1. Page 3, paragraph 3, line 5: change NMDV to NDMVs;

We have changed NMDV to NDMVs

  1. Page 3, last paragraph: change neutrophils to neutrophils'

We have changed neutrophils to neutrophil’s.

  1. Page 4: change MSC-derived EVs to MSC-derived EVs, monocyte derived macrophages to monocyte-derived macrophages.

We have changed to MSC-derived EVs, monocyte derived macrophages to monocyte-derived macrophages

Reviewer 2 Report

   The authors described the role of extracellular vesicles (EVs) in innate immune cell programming in this review. Since recent findings demonstrated that EVs can mediate cellular communication to other cells in local tissue microenvironments and across organ systems, profound understanding about its role and function in innate immune system is important in various pathological aspects. Although there are still many unanswered questions, the authors described those fairly well at the present time. However, in "References" section, several redundancies were seen; i.e., #7 and #74, #43 and #44, #48 and #49, and #105 and #106.

Author Response

We thank the reviewers  and editors for their appraisal and careful consideration of our review manuscript. We are pleased to address all comments raised and include a point-by-point rebuttal below.

The authors described the role of extracellular vesicles (EVs) in innate immune cell programming in this review. Since recent findings demonstrated that EVs can mediate cellular communication to other cells in local tissue microenvironments and across organ systems, profound understanding about its role and function in innate immune system is important in various pathological aspects. Although there are still many unanswered questions, the authors described those fairly well at the present time. However, in "References" section, several redundancies were seen; i.e., #7 and #74, #43 and #44, #48 and #49, and #105 and #106.

We thank the review for their consideration of our manuscript. We have modified our references as suggested by the reviewer.

Round 2

Reviewer 1 Report

The authors have addressed all questions and comments satisfactorily, therefore I support the publication of the revised manuscript.